# Biopriming of Maize Seeds with a Novel Bacterial Strain SH-6 to Enhance Drought Tolerance in South Korea

**DOI:** 10.3390/plants11131674

**Published:** 2022-06-24

**Authors:** Shifa Shaffique, Muhammad Aaqil Khan, Shabir Hussain Wani, Muhammad Imran, Sang-Mo Kang, Anjali Pande, Arjun Adhikari, Eun-Hae Kwon, In-Jung Lee

**Affiliations:** 1Department of Applied Biosciences, Kyungpook National University, Daegu 41566, Korea; shifa.2021@knu.ac.kr (S.S.); m.imran02@yahoo.com (M.I.); kmoya@hanmail.net (S.-M.K.); anjali.pande23@gmail.com (A.P.); arjun@knu.ac.kr (A.A.); eunhaekwon@naver.com (E.-H.K.); 2Center of Biotechnology and Microbiology, University of Peshawar, Peshawar 45000, Pakistan; aqil_bacha@yahoo.com or; 3Mountain Research Center for Field Crops Khudwani, Shere-e-Kashmir University of Agriculture Sciences and Technology, Srinagar 190025, India; shabirhwani@skuastkashmir.ac.in or

**Keywords:** SH-6, seed biopriming, germination, novel isolate

## Abstract

Maize is the third most common cereal crop worldwide, after rice and wheat, and plays a vital role in preventing global hunger crises. Approximately 50% of global crop yields are reduced by drought stress. Bacteria as biostimulants for biopriming can improve yield and enhance sustainable food production. Further, seed biopriming stimulates plant defense mechanisms. In this study, we isolated bacteria from the rhizosphere of *Artemisia* plants from Pohang beach, Daegu, South Korea. Twenty-three isolates were isolated and screened for growth promoting potential. Among them, bacterial isolate SH-6 was selected based on maximum induced tolerance to polyethylene glycol-simulated drought. SH-6 showed ABA concentration = 1.06 ± 0.04 ng/mL, phosphate solubilizing index = 3.7, and sucrose concentration = 0.51 ± 0.13 mg/mL. The novel isolate SH-6 markedly enhanced maize seedling tolerance to oxidative stress owing to the presence of superoxide dismutase, catalase, and ascorbate peroxidase activities in the culture media. Additionally, we quantified and standardized the biopriming effect of SH-6 on maize seeds. SH-6 significantly increased maize seedling drought tolerance by up to 20%, resulting in 80% germination potential. We concluded that the novel bacterium isolate SH-6 (gene accession number (OM757882) is a biostimulant that can improve germination performance under drought stress.

## 1. Introduction

Sustainable agronomy is a new approach to agricultural production that is based on natural resources to preserve the productive capacity of the soil [1,2]. It can also minimize the side-effects of agricultural production on the environment [3,4,5]. Sustainable agricultural practices are becoming increasingly important, as according to estimates, world food production must increase approximately 50% by the year 2050 to ensure food security for the global human population expected by that time [6,7]. The rapid change in the environment owing to climate change and the ever-increasing global demand for food is a catalyst urging us to design and develop new, sustainable farming approaches, such as sustainable agronomy, that rely on the use of biostimulants for seed biopriming [8,9].

Seeds with the potential genetic variations that determine high crop productivity are biologically important for sustainable crop production. Indeed, from this perspective, seeds are the basic prerequisite for food and energy security [10,11]. Specifically, resilient seeds can realize maximum productivity, thereby contributing to sustainable crop production as well [12,13].

The use of a variety of agrochemicals that were previously employed in conventional cropping is now being discouraged because of the resulting environmental hazards and their effects on human health [14,15]. Seed biopriming is an innovative technique that increases germination potential (GP), germination energy (GE), germination rate index (GRI), and seed vigor index (SVI), without harming the ecosystem [16,17], thereby promoting food and energy security [18,19].

Numerous experiments have been conducted to develop and optimize seed biopriming in agricultural production. Improving GP, GE, GRI, and early seedling growth characteristics improves plant stress resistance and overall crop performance [20,21,22]. Arable land is affected by a range of abiotic stress factors, such as drought, and they lead to reduced plant growth and ultimately low crop productivity. Seed biopriming is a promising approach to preventing the negative effects of water deficit on plant growth [8,23,24]. Indeed, plant resistance to abiotic stress can be improved significantly by seed biopriming, as it enhances gene expression related to the plant antioxidant system and plant metabolism to prevent oxidative and growth damage to achieve better yields [25].

Maize (*Zea mays*) is an important staple food produced and consumed globally. Indeed, it is the queen of the cereal foods used by humans and livestock animals [26,27]. This cereal grain is rich in carbohydrates, minerals, vitamins, and fiber. Additionally, it contains folic acid and vitamin A; and traces of magnesium, calcium, phosphorus, manganese, and zinc [28,29]. However, compared to other cereal plants, such as wheat and sorghum, maize requires more water, nitrogen, and phosphorus fertilizers at all stages of growth and development to achieve a high yield [30,31]. Water is crucial to crop production for a range of important reasons, among which, the role it plays as an electron donor in the process of photosynthesis is paramount. Therefore, water scarcity soon becomes a severe abiotic-stress condition [32,33]. Indeed, drought is a major limiting factor reducing maize yields around the world, as it negatively affects plant growth and development at all stages of the crop cycle. Particularly, it reduces seed GP, GE, SVI, and GRI [34,35,36]. Under such conditions, integrated water management and the use of bio fertilizers reportedly increase maize productivity [37,38]. Further, evidence shows that any process that promotes the symbiotic relationship between maize plants and rhizosphere bacteria improves maize productivity [39,40].

The present study was planned to screen bacterial isolates for their bio-efficacy against osmotic stress in agricultural crops. It was hypothesized that rhizosphere around Artemisia plants located near Pohang beach may possess variable numbers of bacteria which endure osmotic regulation, and those could be useful in enhancing abiotic stress tolerance. Therefore, as a preliminary study we investigated their effect in mitigating drought stress in maize through seed biopriming. In this study, we also estimated the effect of seed biopriming on the germination metrics of maize under PEG induced osmotic stress conditions.

## 2. Material and Methods

### 2.1. Isolation and Characterization

Microbes were isolated from the rhizosphere soil of *Artemisia* plants collected from Pohang-si beach Daegu, Republic of Korea (36.0190178 N; 129.343480 E; elevation, 23 m a.s.l.). Two-gram soil samples were mixed in 18 mL of sterilized 0.8% sodium chloride solution, and serial dilutions were prepared by a previously described method with a slight modification [41]. All microbes were isolated and screened for plant-growth-promoting potential. The entire procedure followed was as described by Fischer et al. (2007) [42].

### 2.2. Orange Media Test and Congo Red Assay

Initially, isolates were screened by the orange media test, and the Congo red assay was used to analyze the catalase (CAT) and exo-polysaccharide (EPS) production capabilities of the isolates. The orange media was composed of the reactive orange16 100 ppm and lysogeny agar broth media. Congo red-media assay plates were prepared with LB broth (25 g/L), Congo red (0.8 g/L), agar (1.8%), and sucrose (5%). The prepared media were autoclaved, and assay plates were prepared and incubated for 5–7 days at 30–37 °C. The protocol was as described by Kim et al. (2020) [43], with slight modifications.

### 2.3. Polyethylene Glycol Tolerance Test

Seven concentrations of polyethylene glycol (PEG) 6000: 0%, 5%, 10%, 15%, 20%, 25%, and 30%, were prepared and autoclaved. Culture isolate SH-6 (0.1%) was inoculated to 10 mL of sterilized LB broth media and placed in a shaking incubator at 25–30°C for 24 h. Optical density at 600 nm was measured using a UV spectrophotometer (PG Instruments Ltd., Leicestershire, UK) [44].

### 2.4. Phosphate Solubilizing-Index Assay

The phosphate solubilizing index was measured by preparing the assay plates containing the trypticase soy agar media, Ca_3_(PO_4_)_2_, and agar media. Pure bacterial isolate samples (20 µL) were allowed to grow on the assay plates after they were sealed and placed in an incubator for 3 d. The phosphate solubilizing index was measured using the following equation [45,46].
Phosphate solubilizing index = colony diameter + halo zone/colony diameter

### 2.5. Siderophore Production Assay

For determining the siderophore production, assay plates were prepared using chromeazurol S reagent following the procedure described by Alexander and Zuberer (1991) [47]. Pure bacterial isolate samples (20 µL) were inoculated on the plates, which were then sealed and placed in an incubator for 3–5 d.

### 2.6. Production of Indole Acetic Acid (IAA)

Salkowski’s reagent was prepared by mixing 50 mL of 35% HClO_4_ and 1 mL of 0.5 M FeCl_3_. Then, the reagent was mixed with an equal amount of the bacterial culture. The mixture was vortexed for one min, placed in the dark for 30 min, and the color change was observed. The procedure was as described by Gang et al. (2019) [48,49].

### 2.7. Molecular Characterization

Universal primer 27F (5′-AGAGTTTGATC (AC) TGGCTCAG-3′) was used for molecular characterization [50,51,52]. The obtained nucleotide sequence was searched for similarity in the NCBI website. Mega 10 software was used to construct the phylogenic tree using the neighbor-joining method.

### 2.8. Oxidative Stress Media Test

Bacterial isolates were also tested against multiple oxidative stress. Ascorbate peroxidase was analyzed by measuring absorbance at 290 nm using a spectrophotometer. Superoxide dismutase (SOD) and CAT levels were measured using molecular probes assay-kits (Thermo Fisher, Waltham, MA, USA) [52,53,54].

### 2.9. Quantification of Abscisic Acid (ABA) and Sugar Content in Bacterial Isolate SH-6

Pure LB broth media of SH-9 was centrifuged at 7000× *g* for 12 min, and the concentrated bacterial product was retrieved and filtered. Then, the filtrate was used to determine ABA and sugar contents. ABA was quantified by gas chromatography/mass spectrometry with selected ion monitoring (GC/MS SIM) using an ABA standard [55]. ABA quantification was performed through the method previously published by Khan et al. [43].

For determining the sugar concentration, the filtrate was further filtered through a C18 cartridge (0.45 μm Nylon-66 syringe) and analyzed by high-performance liquid chromatographic (HPLC).

### 2.10. Biopriming Maize Seeds

A completely randomized experiment was designed with eight replications. Maize seeds of the Chodang corn VSC03 were obtained from the Crop Physiology Laboratory in the Department of Applied Biosciences at Kyungpook National University, Daegu, Republic of Korea. Seeds were sterilized with 2.5% sodium hypochlorite for 10 min and then treated with 70% ethanol for 30 s [56]. The new bacterial isolate gene accession OM757882 was grown in LB media and then centrifuged at 10,000 rpm for 8 min to obtain a bacterial pellet. This bacterial pellet was used to bioprime the maize seeds by placing them together with the seeds in petri dishes and dividing them into nine groups: (a) control, (b) microbial isolate only, (c) bacterial solution, (d) 5% PEG 6000, (e) 10% PEG 6000, (f) 15% PEG 6000, (g) 20% PEG 6000, (h) 25% PEG 6000, (i) 30% PEG 6000. An equal number of seeds (*n* = 10) and an equal amount of water (5 mL) were placed in each dish. The plates were sealed and placed in a growth chamber at 28–30/16–18 °C and 60%/80% relative humidity day/night. Germination was recorded at 24 h intervals, and seedling length, seedling biomass, and germination metrics were recorded after 8 d using the following equations [57,58].
Germination percentage (GP) = total number of seeds germinated/total number of seeds × 100;
Seed vigor index (SVI) = average root length + average hypocotyl length × GP,
GE Germination energy (GE) = numbers of germinated seeds on days 4 and 7/total number of seeds × 100, = G4/10 × 100, G7/10 × 100,
$$\mathrm{Germination}\;\mathrm{rate}\;\mathrm{index}\;\mathrm{GRI}=\left[\frac{G1}{1}\right]+\left[\frac{G2}{2}\right]+\left[\frac{G3}{3}\right]+\left[\frac{G4}{4}\right]+\left[\frac{G5}{5}\right]+\left[\frac{G6}{6}\right]+\left[\frac{G7}{7}\right]+\left[\frac{G8}{8}\right]$$

### 2.11. Early Seedling Metrics

Seedling length (cm), average root length (cm), average hypocotyl length (cm), fresh weight (mg), and dry biomass (mg) were measured [59].

## 3. Statistical Analysis

Molecular characterization and construction of phylogenic tree were performed using MEGA 10 software (version 7.222). All experiments were replicated five times. Graph pad Prism version 5.8 was used to perform statistical analysis. The mean values were evaluated by using DMRT analysis, SAS 9.1 (Duncan’s multiple range test) at *p* ≤ 0.05 via students T test.

## 4. Results

### 4.1. Molecular Characterization Assay

The obtained nucleotide sequences were searched for similarity on NCBI. The results showed the new SH-6 isolate has similarity with *Enterobacter ludwigii*. Using Mega 10 software, a phylogenic tree was constructed by the neighbor-joining method. The sequences were submitted to Genebank. They were all identified under a unique and distinct gene accession number, i.e., OM757882. These results clearly proved that SH-6 is a novel bacterial isolate, as shown in Figure 1.

### 4.2. Analysis of Siderophore, Indole Acetic Acid, and Exopolysacchrides

The novel isolate was also checked for production of the siderophore, EPS, and IAA, and for phosphate solubilizing index. The results showed that this isolate produced significant amounts of EPS, siderophore, and IAA, as shown in Figure 2. The isolate was also screened for phosphate solubilizing index, and results showed it had a high phosphate solubilization index of 3.69 ± 0.30 cm with a 1.59 ± 0.11 cm colony diameter and 4.29 ± 0.25 cm halo zone diameter.

### 4.3. Drought Tolerance Assay Results

The bacterial isolate was further tested for drought tolerance in PEG 6000 media. Various concentrations of PEG 6000 were prepared for culturing the isolate. The results showed that the isolate effectively tolerated up to 20% PEG 6000 but was severely affected by 25% PEG 6000, as shown in Figure 3.

### 4.4. Sucrose and Abscisic Acid Analysis Results

The results of ABA quantification by GC-MS showed that SH-6 contained significant amounts of ABA (1.06 ± 0.05 ng/mL). Similarly, free sucrose, as measured by HPLC, was also significantly higher (0.58 ± 0.14 mg/mL) than in the control group (0.04 ± 0.004 mg/mL) (Figure 4).

### 4.5. Oxidative Stress Tolerance Response

The results of the oxidative stress tolerance test are shown in Figure 5. SH-6 showed different levels of antioxidant enzyme activities representing the antioxidant system of the bacterial isolate against drought stress. Particularly, CAT activity was strikingly high under stress.

### 4.6. Effect of Seed Biopriming on Germination under Drought Tolerance

Maize seeds of the Chodang VSC03 variety (Asian Gardner) were bioprimed with the novel bacterial isolate SH-6 to test its effects on germination metrics and early seedling growth characteristics. The results show that maize seeds bioprimed with the novel bacterial isolate SH-6 tolerated significantly a degree of drought stress induced by up to 10% PEG 6000; tolerance to 15% and 20% PEG 6000 was moderate, but 25% PEG 6000 severely affected germination metrics. The bacterial isolate enhanced GP, GE, GRI, and SVI. Specifically, GP reached 80% of the control value under the 20% PEG 6000-induced water stress. Values for SVI (425 ± 0.34) are comparable with those recorded for the control group. The results are shown in Figure 6. Seed biopriming enhanced average hypocotyl and root length in maize seeds as shown in Table 1. Furthermore, GRI, which is an indicator of germination over a time, was also enhanced by seed biopriming.

## 5. Discussion

Maize is the third most important cereal crop consumed as a staple food globally, after rice and wheat [60,61]. It is cultivated as a cereal grain throughout the world. It is known as the queen of the cereals because of its genetic yield potential [62]. Maize plants require large amounts of water for successful growth and development to achieve their potential yields [63]. Maize is gaining importance worldwide because of the nutrient content available in the grains. The agriculture industry has to face dual pressure: environmental stresses such as drought stress, and meeting the needs of the population [64].

Among environmental stresses, drought is the leading stress that affects plant productivity. The agricultural industry frequently faces environmental threats because of the sessile nature of plants [65,66]. Unfortunately, 35–40% of our land is semi-arid, whereby maize crops are constantly challenged with drought stress, which affects plant growth and development at every stage, from germination to maturity, thereby causing severe crop-yield losses [67,68]. Conventional strategies fail to meet the needs of the agriculture industry because of their wide range of side-effects [69]. To counter these negative effects of drought [70,71,72], seed biopriming is a promising approach, as it effectively enhances the plants’ drought tolerance, improves germination, and ultimately, improves yield [27,73,74]. Further, as a feasible, ecofriendly approach, biopriming is an alternative to the conventional approaches that rely on agrochemicals [75,76,77]. Increasing public awareness on the environmental and human health hazards entailed by such approaches has led to their decline and to increasing interest and acceptance of alternative strategies for coping with the negative effects of abiotic stress on agricultural production [16,21,78].

Drought stress is a natural ecological burden on agriculture production. Beneficial microbes play an important role in providing the resistance and adaptation of plants to osmotic stress, and will have a key role in providing food and energy security in future. The results presented here demonstrated that biopriming enhance plant growth and development during drought stress [79,80]. Plant-growth-promoting rhizobacteria have been examined for their ability to combat drought stress in nature. They have the ability to colonize plant cells and establish a long mutual symbiotic relationship [81,82]. They not only provide the safety to plants, but also are a source of energy. In plants, the microbial involvement in drought tolerance is attributed to the production of the phytohormones such as indole acetic acid, absicis acid, sucrose, and oxidative stress tolerance, as shown in Figure 7 [83,84,85].

In the present study, rhizobacteria were isolated from Pohang beach from Artemisia plants and screened for growth promoting characteristics. From 23 bacterial isolates, one competent bacterial isolate SH-6, was selected. SH-6 produced significant amounts of IAA, a natural auxin that promotes cellular plant growth [86,87]. Furthermore, SH-6 produced EPS, which plays a critical role in the formation of a biofilm that is very important for attachment of the microbe to the surface of plant roots. The resulting plant-microbial interaction also contributes to the availability of siderophore, which in turn contributes to the availability of iron for plant cells under the stress condition [88,89]. The novel isolate SH-6 was also analyzed for the production of ABA, another important phytohormone in plant growth and stress responses. Specifically, during water stress, ABA is a signaling molecule responsible for stomatal closure to minimize transpirational water loss [90,91]. Thus, ABA also enhances drought resistance and improves early seedling metrics in maize. HPLC analysis showed that SH-6 contained significant (1.08 ± 0.05 ng/mL) amounts of ABA. Additionally, GC-MS results showed that SH-6 accumulated significant amounts (0.58 ± 0.14 mg/mL) of soluble sugars. Particularly, sucrose is very important in plant nutrition. After preliminary screening, the isolate was selected for in vitro seed biopriming to see its effects on germination metrics. The results showed that the isolate effectively and significantly preserved germination metrics with an up to 20% induced osmotic stress level. After 20%, there was inhibited germination, and the isolate did not tolerate the drought stress significantly, resulting in 16.96 ± 0.09 mg, 1.33 ± 0.06 mg fresh and dry biomass, respectively. This may be due to several mechanisms, i.e., morphological, biochemical, and metabolic [92,93,94].

When plants are exposed to drought stress, reactive oxygen species accumulate, resulting in the degradation of cellular proteins and loss of turgor due to membrane damage caused by membrane lipid peroxidation, which in turn leads to cytoplasmic leakage [95,96]. Biopriming improves the defensive mechanisms, selective absorption, membrane stability, and stress tolerance [97]. The results summarized here in proved that the novel isolate SH-6 tolerated oxidative stress well owing to high SOD, CAT, and APX activities, and improved the plant’s antioxidant system. As for germination metrics, the data showed that the maximum level of stress tolerance of maize seeds was observed under 20% PEG 6000 stress, whereas water-deficit levels above that induced by 20% PEG 6000 negatively affect seed germination metrics. This might be due to membrane stability and the defensive mechanism of the seed biopriming approach. Maize seeds tolerated stress up to 20%, indicating that seed biopriming only allows selective absorption. It likely restricted the polyethylene glycol from entering plant cells, ultimately inhibiting the drought stress.

The germination metrics, i.e., seed vigor index, germination energy, and germination rate index, decreased with increasing stress level. Inoculation with a bacterial strain can rescue the stress tolerance by only up to 20%, as shown in the Figure 6. After that, there is inhibition of the germination, which clearly indicates that SH-6 can tolerate stress only up to 20%. The results clearly demonstrate that the new isolate can serve as a seed biopriming agent for drought stress tolerance and enhancing the germination metrics.

Phytohormones play an important role in the mitigation of the stress. The bacterial isolate produced absicis acid significantly as compared to the control group. Absicis acid is known as the stress hormone which is involved in stress tolerance. When a plant is exposed to drought stress, ABA is released, and it causes the closure of the stomata to reserve the water from evaporation [98,99]. In the present study, the SH-6 improved the stress tolerance that may be due to production of ABA by a bacterial isolate.

The present study confirmed the growth promoting characteristics and improved germination potential under drought stress with the production of phytohormones, sucrose, exopolysacchrides, and siderophore. Further intensive research is needed to evaluate the potential of SH-6 under open agronomic management.

## 6. Conclusions

Bacteria are natural biostimulants that have the potential to improve crop seeds’ germination metrics under drought and improve plants’ cellular responses to stress. The novel bacterial isolate SH-6 improved drought tolerance under drought stress, as simulated by the addition of as much as 20% PEG 6000 to the germination media, and it can be used as a biostimulant to increase maize yield. SH-6 contributed positively to the germination potential and seed vigor index. The application of bacteria as seed biopriming agents is an inexpensive and effective approach for the mitigation of drought stress effects. Furthermore, they have a positive effect on the plant’s antioxidant system under conditions of drought stress. To conclude, the application of rhizosphere bacterial isolate SH-6 improved germination of maize seeds under drought conditions.

## Figures and Tables

**Figure 1 plants-11-01674-f001:**
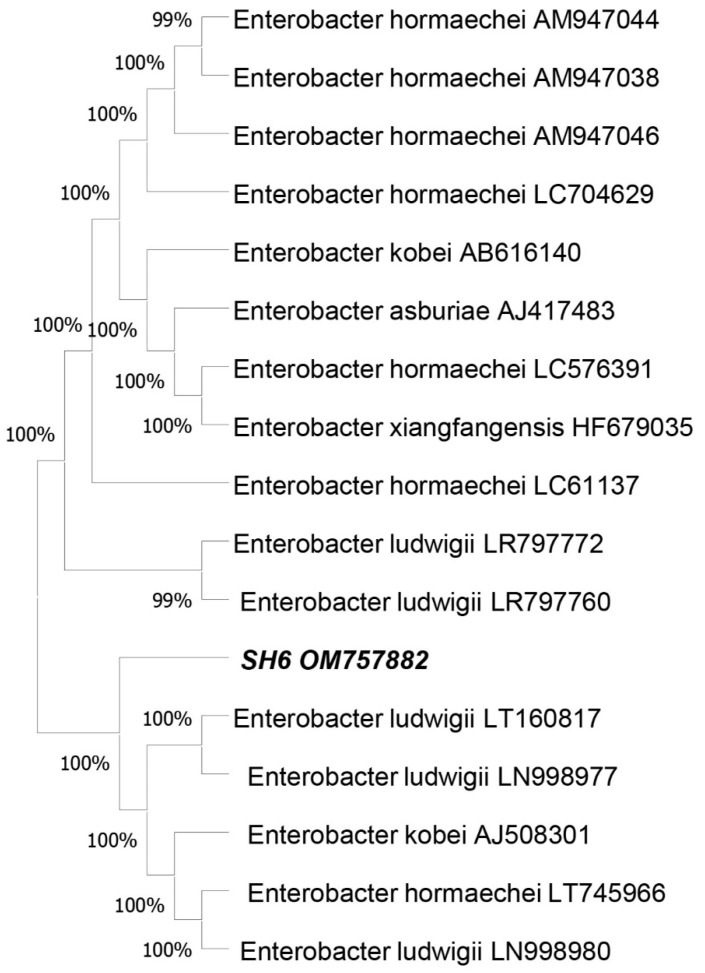
Phylogenic tree of SH-6 16S rRNA with 16S rRNA of other strains of *Enterobacter ludwigii*. The phylogenetic tree was constructed using the MEGA 10software.

**Figure 2 plants-11-01674-f002:**
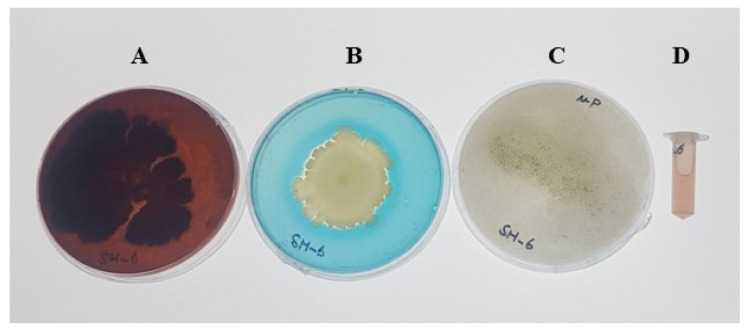
(**A**) Exopolysaccharide, (**B**) phosphate solubilization, (**C**) siderophore production, and (**D**) IAA production by the new rhizosphere bacterial isolate SH-6.

**Figure 3 plants-11-01674-f003:**
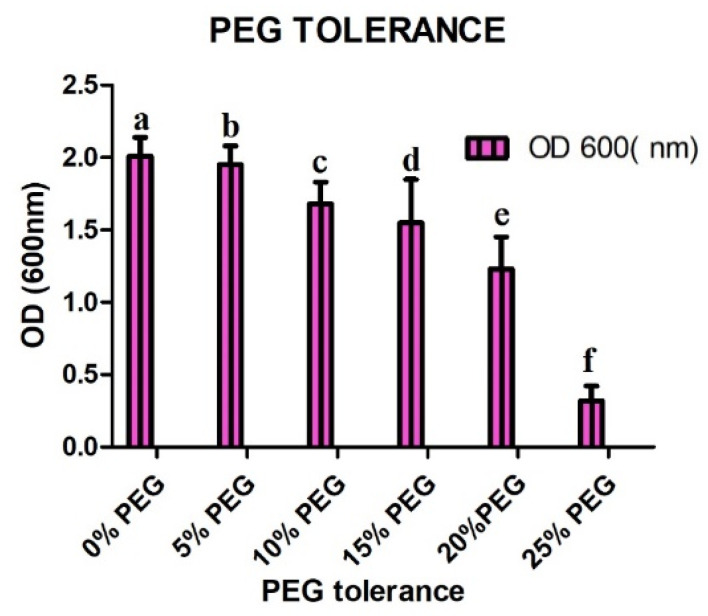
Growth of SH-6 under different concentrations of PEG 6000. The latter on each bar is the significant different between treatment at *p* ≤ 0.5, where ND = not detected. The error bar represents the standard error among the replicates.

**Figure 4 plants-11-01674-f004:**
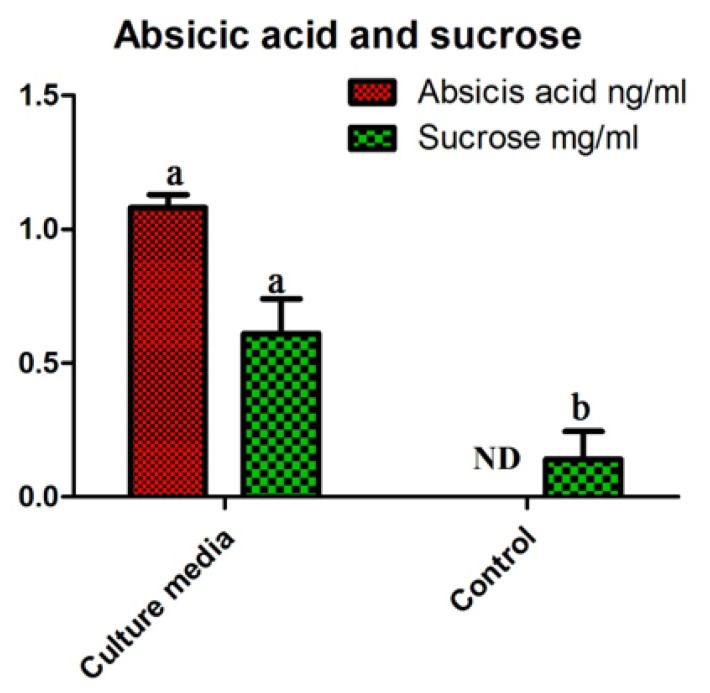
ABA and sucrose quantification in isolate SH-6 cultured in LB broth media. Each data point is the mean of 5 replicates. The latter on each bar is the significant different between treatment at *p* ≤ 0.5, where ND = not detected. The error bar represents the standard error among the replicates.

**Figure 5 plants-11-01674-f005:**
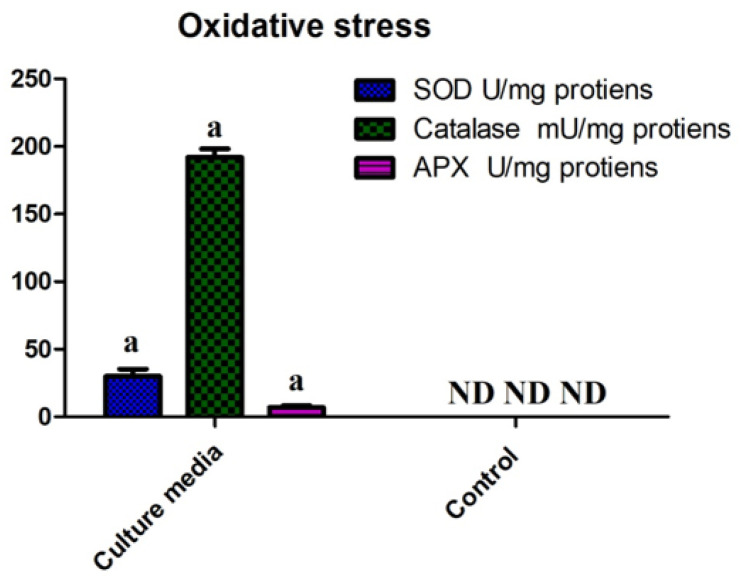
Oxidative stress tolerance activities of SH-6. Each data point is the mean of 5 replicates. The latter on each bar is the significant different between treatment at *p* ≤ 0.5, where ND = not detected. The error bar represents the standard error among the replicates.

**Figure 6 plants-11-01674-f006:**
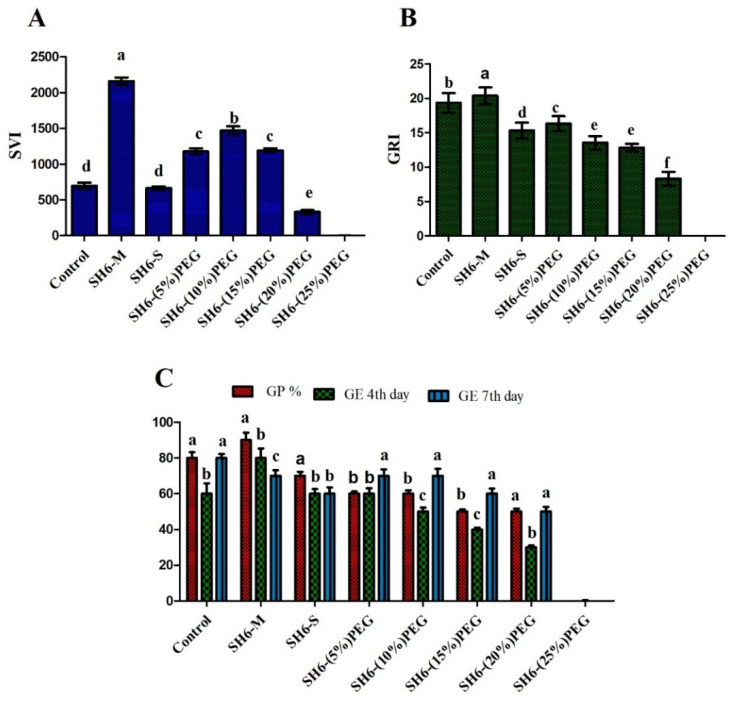
Germination metrics. (**A**) SVI, (**B**) GRI and (**C**) GP and GE. Each data point is the mean of 5 replicates. The latter on each bar is the significant different between treatment at *p* ≤ 0.5, where ND = not detected. The error bar represents the standard error among the replicates.

**Figure 7 plants-11-01674-f007:**
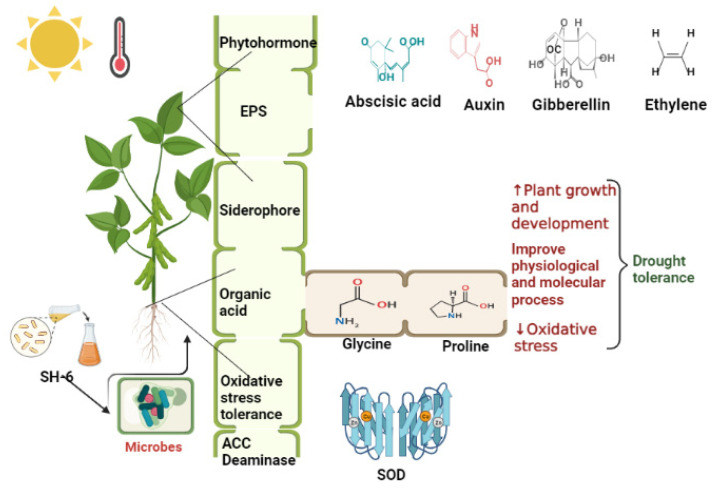
Microbial mitigation of drought stress.

**Table 1 plants-11-01674-t001:** Seedling characteristics and biomass at day 7. The letters indicate the significant difference between treatment at *p* ≤ 0.5.

Treatment Groups	Fresh Biomass (mg)	Dry Biomass (mg)	Root Length cm	Hypocotyl Length cm
SH-6-M	27.75 ± 0.06 a	1.93 ± 0.08 a	14 ± 0.05 a	5.5 ± 0.01 a
SH-6-S	20.96 ± 0.04 d	1.91 ± 0.05 b	0.5 ± 0.02 g	4.2 ± 0.01 d
SH-6—(5%) PEG 6000	24.82 ± 0.02 b	1.88 ± 0.07 c	11 ± 0.02 b	5.3 ± 0.03 b
SH-6—(10%) PEG 6000	21.27 ± 0.07 d	1.78 ± 0.07 d	10.7 ± 0.08 c	5.2 ± 0.02 c
SH-6—(15%) PEG 6000	18.46 ± 0.08 e	1.25 ± 0.03 f	5.4 ± 0.01 e	3.2 ± 0.02 e
SH-6—(20%) PEG 6000	16.96 ± 0.09 f	1.33 ± 0.06 g	7.9 ± 0.04 e	1.9 ± 0.04 f
SH-6—(25%) PEG 6000	2.56 ± 0.09 g	1.27 ± 0.08 h	1.02 ± 0.04 f	1.01 ± 0.06 g
Control	21.34 ± 0.32 c	1.62 ± 0.18 e	9.69 ± 0.03 d	4.55 ± 0.02 c

## Data Availability

Not applicable.

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
