# Peer review of "Biopriming of Maize Seeds with a Novel Bacterial Strain SH-6 to Enhance Drought Tolerance in South Korea"

_plants, 2022, doi:10.3390/plants11131674_

Round 1
Reviewer 1 Report
Drought stress is one of the major limiting factors in agriculture production. Beneficial microbes play an important role in providing the resistance and adaptation of plants to osmotic stress and have a key role in providing food and energy security in the future. Even though the authors provide plenty of information to show that novel bacterial isolate SH-6 improved drought tolerance, some of them look interesting, I still suggest rejecting this version due to the following reasons.
The manuscript looks more like a draft, not well prepared from the figures presented and results in the texts.
1. The authors said, “Twenty-three isolates were tested for growth promoting potential.” The figures presentation did not show any.
2. There is no statistical analysis at all for all the bar figures.
3. The results presentation needs a lot of work, difficult to follow.
4. What is the control of Figure 5?
5. I did not see any information about Figure 7, what is its purpose?
6. More details are as follows:
Line 59: Is there any evidence or reference to make the authors say that “it is the queen of the cereal foods used by humans and livestock animals”?
Line 85: “after a slight modification”, what is the modification? The same question is in line 95.
Line 139-140: “rice seeds”? I think the authors mean “maize seeds”.
Line 165: “Statistical analysis” should be in the Material and methods section, not the Results section.
Line 166-168: What the authors present here is not about the statistical analysis, what test did they use?
Line 172: What are other sequences for the phylogenic tree?
Line 180-181: “The results showed that this isolate produced significant amounts of EPS, siderophore, and IAA, as shown in Figure 2.” Compare to what?
Line 227-228: “35%”, reference?
Line 231-233: “seed biopriming is a promising approach, as it effectively enhances the plant drought tolerance, improves germination, and ultimately, maize yield.” Any example or reference?
Author Response
Dear reviewer,
thankyou so much for your valuable comments.Please find the attached paper

Reviewer 2 Report
Dear Authors,
The title of the article sounds interesting. Also, performed analyses look interesting. Article focuses on the important issue of how drought affects on crop yields
The experiments are reproducible. Results presented in a clear authoritative reliable form.
Presented conclusions are the result of the analysis carried out
However:
Recommendations for authors:
1. Please check the correctness of writing the name- upper case, lower case
Mountain research for field crops khudwani, shere-e- Kashmir University of agriculture sciences and technology Srinagar, jamu and Kashmir
2. Graph captions – standardize- upper case, lower case; eg. Fig 6?
3. Figure 1 - disproportionately large
4. Have other statistical analyses been performed? No information on statistical differences. In my opinion, operating only on averages and errors is not very representative
5. In my opinion, the discussion is a bit too short
6. One year study? Was this done in any series or just replicates?
7. Please read the MS once more and correct any minor shortcomings, e.g. punctuation, etc.
Author Response
Dear reviewer,
thankyou for your valuable comments

Reviewer 3 Report
Dear authors!
The manuscript is devoted to the actual topic of increasing the drought resistance of food crops with the help of microorganisms. However, there are many serious comments to the manuscript, which significantly reduce its quality.
1. The manuscript (especially the References section) is not written according to the journal template, which makes it difficult to review.
2. The English language of the authors needs a lot of improvement.
3. Keywords do not give an idea of ​​what the text is talking about.
4. At the end of the Introduction section, the authors write (lines 74-76): "We conducted this study to isolate and characterize plant growth-promoting bacteria (PGPB) and to determine the effect of the obtained bacterial isolates on the drought tolerance of maize plants". However, both the title of the manuscript and its text refer to only one bacterial isolate, SH-6. It is necessary to formulate the hypothesis and the purpose of the study more clearly.
5. The authors included an unnumbered Graphical abstract in the manuscript (page 3). Why did they do it? Is there a need for this picture?
6. Lines 101-102. "Based on preliminary screening results, bacterial isolate SH-6 was selected for further study [42]". Does this mean that the resistance of the SH-6 isolate to PEG has already been studied by the authors of the article [42]? If so, why do the authors include in their manuscript what has already been investigated by other scientists (section 2.3 Polyethylene glycol tolerance test and Figure 3)? Attempts to find the article [42] on the Internet were unsuccessful.
7. Lines 122-123. "The obtained nucleotide sequences were searched for similarity in the NCBI website", line 170 "The obtained nucleotide sequences were searched for similarity on NCBI". However, only one nucleotide sequence number is indicated in the manuscript (strain SH-6).
8. Why biopriming rice seeds are described in Section 2.10 when the manuscript is about increasing drought tolerance in maize?
9. Lines 162-163. "Seedling length (cm), average root length (cm), average hypocotyl length (cm), fresh weight (mg), and dry biomass (mg) were measured". Yet, in the Results section, these data are absent, there are only photographs of corn seedlings (Fig. 7).
8. Section 3.1 should be moved to Materials and Methods.
9. What is the percentage of similarity of strain SH-6 with representatives of the genus Enterobacter? If the authors are not sure about the species of the strain, then perhaps it should be designated as Enterobacter sp. SH-6.
10. The phylogenetic tree is not informative. Other members of the Enterobacter genus should be included. When constructing a dendrogram, it is necessary to indicate the reliability of branching.
11. Table 1 is very small and uninformative. The data can be transferred to the text of the manuscript.
12. Line 248-249. “…SH-6 produced significant amounts of IAA…”. There are no data on the number of IAA in the manuscript.
13. Section Conclusion (lines 280-281). "...and it (SH-6) can be used as a biofertilizer to increase maize yield." To make such assumptions, the authors should conduct additional laboratory and field tests.
14. Figures 3-6 do not indicate the significance of the differences.
Author Response
thankyou for your valuable comments

Round 2
Reviewer 1 Report
The reviewer's comments and questions were well addressed in the revision. This reviewer thinks this MS could be accepted in its present form.
Reviewer 3 Report
Dear authors!
Thank you for your answers.